# An unclean slate, discrepancies between food input and recovered protein signal from experimental foodcrusts

Joannes Adrianus Antonius Dekker [1,2,3*], Richard Hagan[3], Matthew Collins[2,4], Jessica Hendy[3]

1 Department of Life Sciences and Systems Biology, University of Turin, Turin, Italy, 2 Section for Geobiology, Globe Institute, University of Copenhagen, Copenhagen, Denmark, 3 BioArCh, Department of Archaeology, University of York, York, United Kingdom, 4 McDonald Institute for Archaeological Research, University of Cambridge, United Kingdom

* jan@palaeome.org

## Abstract

Organic residues are a rich source of biomolecular information on ancient diets. In particular, foodcrusts, charred residues on ceramics, are commonly analysed for their lipid content and to a lesser extent protein in order to identify foods, culinary practices and material culture use in past populations. However, the composition of foodcrusts and the factors behind their formation are not well understood. Here we analyse proteomic data (available via ProteomeXchange with identifier PXD059930) from foodcrusts made using a series of mixtures of protein- (salmon flesh), lipid- (beef fat) and carbohydrate-rich (beetroot) foods to investigate the relationship between the biomolecular composition of the input and the recovered signal using conventional methods applied to archaeological material. Additionally, using 3D modelling we quantify the volume of foodcrust generated by different ingredient combinations The results highlight biases in the data obtained in the analyses of organic residues both in terms of identified resources reflecting the cooked foodstuffs, e.g., an overrepresentation of fish proteins, as well as with regards to the abundance of foodcrust, for example mixtures of only salmon and beef fat resulted in relatively small amounts of foodcrust, and suggest caution in interpreting the composition of residues formed from complex mixtures of foodstuffs.

## Introduction

Food residues on or in ceramic vessels are a popular target for studies on ancient diets and foodways. Their potential for revealing components of the diets of past people is demonstrated by studies on the adoption of pottery by hunter-gatherers [1] and continuity in diet under political regime change [2]. Organic residues on archaeological vessels can manifest in different ways, including calcified deposits [3],

**Data availability statement:** A full submission of the acquired mass spectrometry data has been made to the ProteomeXchange Consortium via the PRIDE 64 partner repository with the dataset identifier PXD059930 and https://doi.org/10.6019/PXD059930. Additional metadata of the MetaMorpheus database search and the 3D model object files of the experimental ceramic vessels used in the cooking experiment can be found on Zenodo under the following DOI: https://doi.org/10.5281/zenodo.15874520.

**Funding:** This project was made possible thanks to funding from the European Union's Horizon 2020 research and innovation programme under the Marie Skłodowska-Curie grant agreement No 956351. The funders had no role in study design, data collection, analysis, decision to publish or preparation of the manuscript.

**Competing interests:** The authors have declared that no competing interests exist.

charred foodcrusts (hereafter foodcrusts) [4] and organic residues entrapped within the ceramic matrix [5], and more rarely, organic remains preserved on the surface of vessels through arid, frozen or waterlogged conditions [6–8]. These three different types of residues are formed by different processes and are affected differently by diagenesis. Of the three types of residue, we focus on foodcrusts in this study. Foodcrusts have so far been mostly analysed for their lipid content (including fats, oils and waxes). Lipid analysis is robust and has been successfully applied to materials from a wide range of preservation conditions spanning the globe and stretching back as far as at least 12 000 years ago [2,9–11]. For example, the study of lipids has been successful in studying the emergence and spread of the use of dairy products [12–14] as well as the continuity in the widespread use of marine resources through the Mesolithic into the Neolithic [15–18]. However, there are still areas for improvement. The results of compound-specific isotope analysis can be confounded by the mixture of multiple resources and the results are likely biased towards lipid-rich foodstuffs compared to lipid-poor foods, such as most plants [19].

The presence of plants in foodcrusts is best investigated using SEM (scanning electron microscopy) [20,21], revealing, for example, the presence of acorns in ceramic vessels in the Neolithic Netherlands [22]. Animal remains, although difficult to taxonomically resolve, can also be observed using SEM [20].

In theory, the disadvantages of both lipid analysis and SEM could be circumvented by protein analysis. As proteins are encoded by DNA, they inherently carry phylogenetic information, although the taxonomic specificity of a protein varies greatly between proteins. Beyond their use for taxonomic identification, some proteins can be used to identify the tissues that were cooked, as some proteins are abundant in or exclusive to specific tissues. For example, the frequently observed protein beta-lactoglobulin (BLG) is specific to dairy [23–26], and is not synthesised by humans at all. Additionally, as proteomic taxonomic identification is based on the presence of diagnostic peptide sequences, protein analysis should be able to distinguish between the different components of a food mixture. However, the application of protein analysis to foodcrusts is in its infancy [4,7,27,28] While some success has been observed [4,29], biases and challenges in this method have been noted [12]. For example, the cooking and burning process of foodcrust formation likely impacts the numbers of food proteins, which may or may not be taxonomically diagnostic. Furthermore, in the case of analysing mixtures of ingredients, it is unknown whether the foodcrust preserves an accurate reflection of the input food proteins. Protein analysis will of course favour protein-rich foods, but other foodstuffs less enriched in protein, such as plants, still contain characteristic proteins and have been identified in organic residues [3]. Whether there are additional factors beyond the initial protein content determining the likelihood of a particular protein's survival is unclear. A recent study on protein incorporation and survival in the ceramic matrix and foodcrust tested the influence of several physico-chemical properties, but beyond a slight favour towards hydrophobic peptides, there were no clear universal trends in protein survival [30]. Beyond these general limitations, not much is known about the challenges of applying protein analysis to organic residues nor about how the theoretical advantages translate into reality.

This study will focus on foodcrusts and aims to clarify the strengths and limitations of protein analysis using experimental foodcrusts made of a mixture of protein-, lipid- and carbohydrate-rich foods. Specifically, we will investigate how mixtures of different foodstuffs might influence the volume of foodcrust formed and the recoverable proteome. In particular, we will examine the degree to which the 'extractome', the suite of recovered proteins [31], reflects the original input food mixture, in order to highlight the potential biases involved in the identification of foodstuffs in archaeological material. Additionally, this study aims to map how repeated cooking events and burial influence the extractome's composition and the volume of generated foodcrust, adding to our understanding of the diagenesis of proteins in charred organic residues.

In order to maximise the relevance of the experiment's results for archaeological samples the choice of foodstuffs is vital. As this experiment is part of a larger project focused on Mesolithic foodways in Denmark, we selected foodstuffs that would be similar to those utilised by late Danish hunter-gatherers. Secondly, as the three commonly used methods, i.e., lipid analysis, proteomics and SEM, to study organic residues all favour a different class of biomolecule we decided to select one lipid-rich, one protein-rich and one carbohydrate-rich foodstuff to mix. Using foodstuffs with vastly different nutritional compositions facilitates visualising the biases towards either carbohydrates, proteins or lipids. Therefore, care was taken to ensure that the relative proportions of the wet weights of the carbohydrate-rich, protein-rich and lipid-rich foodstuffs matched as closely as possible to the ratio of carbohydrates:proteins:lipids in the total mixture (e.g., a mixture consisting of 20% of the protein-rich food would also contain 20% proteins). Perfect symmetry between the ratios of wet weights and the molecular categories was not possible, especially not when limited to foodstuffs likely available in Mesolithic Denmark, but the closest match was obtained using salmon, beef fat and beetroot (SI 1). However, we acknowledge that this combination of foodstuffs underrepresents the carbohydrate component in particular.

Beetroot was chosen as the carbohydrate-rich food due to the prevalence of its wild ancestor *Beta vulgaris maritima* at Mesolithic sites such as Tybrind Vig and Møllegabet II [32,33], as well as the fact that it has a relatively well-annotated reference proteome. Protein identification is heavily dependent on the quality and coverage of reference proteomes. Low-quality reference proteomes only contain a fraction of the total proteome of a species and any proteins not included cannot be identified. To highlight the magnitude of this issue, the median number of proteins listed on UniProt [34] for 29 plant species found at Mesolithic northern European sites [32,35–38] is 120 proteins (SI 2), in contrast to the 7679 available for *Beta vulgaris* or the 130 673 for *Triticum aestivum* (wheat), a high-quality reference proteome.

We selected beef fat and salmon because the exploitation of aurochs (*Bos primigenius*) and salmon (*Salmo salar*) is well attested at a number of Danish late Mesolithic sites [36,39–42]. Additionally, salmon has a well-annotated reference proteome covering 90% of its total proteome. Available reference data for aurochs is much more limited, but due to the short evolutionary distance between them, domestic cattle (*Bos taurus*), which has a 97% complete reference proteome, was used.

By varying these three ingredients in regular increments we are able to establish the presence of any biases favouring or disadvantaging the recovery of proteins from any of the foods. In turn, this new insight on the presence of biases will facilitate the interpretation of proteins identified in archaeological foodcrusts and thus our understanding of past foodways. The analyses presented here demonstrate that, as expected, there is a bias towards the recovery of proteins from protein-rich foods and that it is likely that not all ingredients can be identified from a food mixture.

## Materials

The ceramic vessels used in the cooking experiments were purchased from Potted History, a company specialising in historical ceramic replicas. The specifications of the ceramics are in line with the pottery used in the experiments by Bondetti et al. [43]: they were hand thrown with the 'standard red' clay ($Al_2O_3$, 22.78; $Fe_2O_3$, 7.37; CaO, 0.57; MgO, 0.86; $K_2O$, 1.6; and $Na_2O$, 0.1), had a diameter of ~10 cm, a wall thickness of ~0.5 cm and were fired at 700 °C. The foodstuffs used in the experiments were all obtained from local commercial vendors; the beetroot was purchased from Morrisons, the beef fat from M&K Quality Butchers, York, and the salmon was purchased online from LondonGrocery.net ([https://](https://)

). As what is often sold as 'salmon' can actually belong to a variety of species and even genera, the purchased specimens were examined for morphological characteristics [44] to ensure that they were in fact *Salmo salar*.

## Methods

### Cooking experimental protocol

In order to test how different combinations of carbohydrate-, protein- and lipid-rich foodstuffs influence foodcrust formation it was decided to mix the three selected foodstuffs, beetroot, salmon and beef fat, in all possible combinations using 20% increments, which results in 21 sample conditions (Fig 1). To each of these unique combinations one ceramic vessel was assigned. Additionally, we included one ceramic vessel as a negative control, which would only be filled with water during cooking. No samples were taken of the negative control for proteomic analysis, as indeed no foodcrust formed on the water filled vessel. Instead, an extraction blank was included in the protein extraction process to serve as negative control.

Before the first cooking event, all pots were soaked in tap water at room temperature for 15 minutes in order to prevent cracking when heated. The vessels were then filled with the ingredients according to their experimental conditions to a total mass of 100 g. Additionally, 100 mL of distilled water was added to each vessel. The filled vessels were wrapped in aluminium foil, leaving the top open, and placed for an hour in an oven (Binder) preheated to 270°C. The aluminium foil was intended to hold the pots and their contents in place in the event of the ceramic fracturing, as pilot tests had shown this to be a possibility. The cooking temperature of 270°C was selected based on previous research that found that APAAs (ω-(O-alkylphenyl) alkanoic acids) often found in archaeological foodcrusts, formed after heating for one hour at 270°C [43]. Indirect heating using a convection oven was chosen in order to ensure a stable and even temperature for all vessels and throughout the cooking event. Direct heating is likely more representative of cooking during Mesolithic times, but introduces substantial variability in the precise temperatures the food is exposed to both during a cooking event as well as

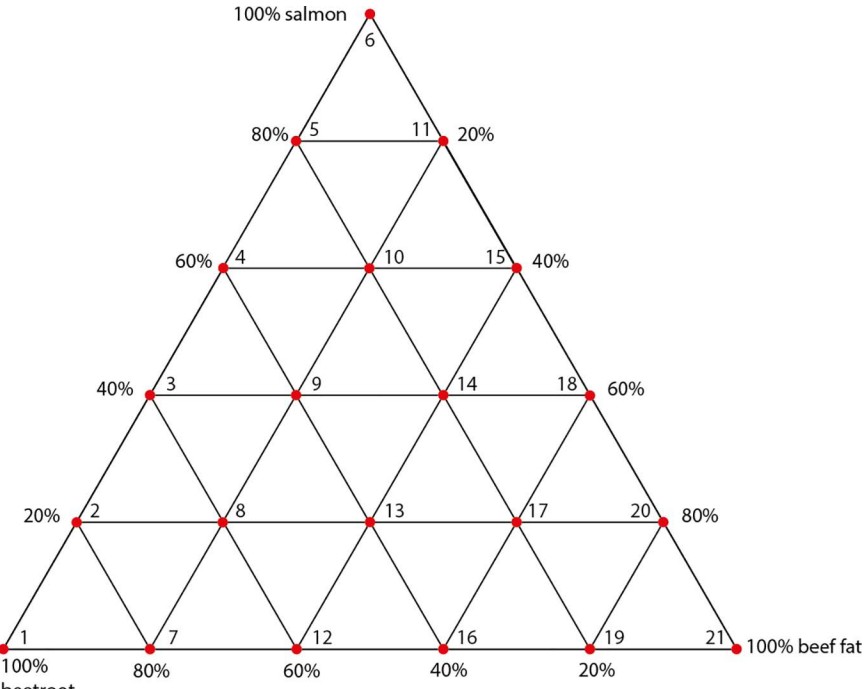

**Fig 1. Overview of experimental conditions.** Each red node indicates an experimental vessel.

between different cooking events and between different vessels within the same cooking event. As we expected temperature to play an important role in the formation of foodcrust, it was preferred to keep it as stable as possible.

After cooking for one hour, the vessels were removed from the oven and allowed to cool down at room temperature. The cooled foodstuffs were then removed from the vessels as we were interested in multiple cooking events. Care was taken to only remove the foodstuffs and to leave the foodcrusts intact. Charred remains that adhered to the ceramic wall, but were macroscopically still recognisable as food, were removed as well. Fig 2 provides some examples of foodcrusts that were formed after the final cooking event.

After removing the food, the vessels were upturned for roughly an hour to allow any liquid to drip out. Any fat that consolidated on a vessel's rim during this period was not considered foodcrust and removed. For subsequent cooking episodes, the vessels were no longer soaked in water. Instead, the vessels were immediately placed in the oven after filling them with their ingredients and 100 mL water. In total nine cooking events were performed for each pot and foodcrust samples were taken after the first, fifth and ninth cooking episodes. Foodcrust could form on the rim, body and bottom of the vessel depending on the food input, but the rim of the vessel was the only location that always contained sufficient foodcrust for sampling, and hence foodcrust was consistently sampled from the rim. Where possible samples were still taken from the bottom and body of the vessel.

After the ninth cooking episode, the vessels were buried with their foodcrusts in the York Experimental Archaeological Research (YEAR) Centre for 52 days at a depth of 50 cm. The sediment the vessels were buried in was sandy and the soil

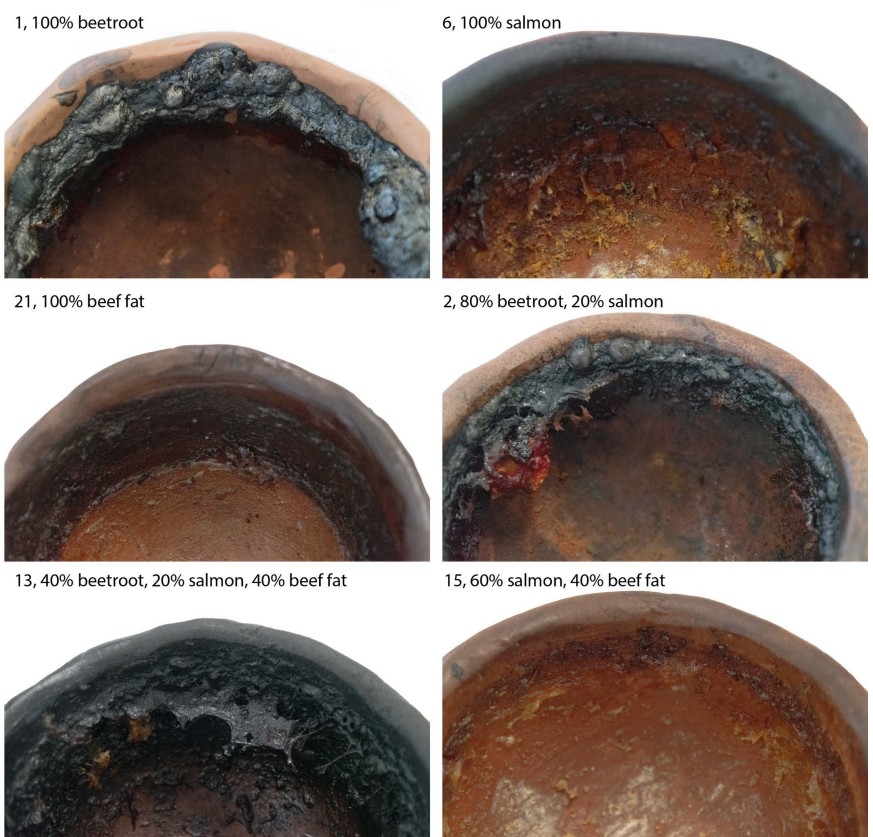

**Fig 2. Foodcrusts formed after the ninth cooking event.** The number denotes the experimental of each vessel and corresponds to Fig 1.

pH of the same area was previously measured to be 7.16 [43]. For the months of August and September, during which the burial experiment was carried out, the Met Office National Climate Information Centre reported a mean monthly air temperature of 16.9 °C and 13.5 °C for North England respectively [45]. The vessels were wrapped loosely in Galvanised wire netting (13 mm mesh size) to prevent bioturbation from any larger creatures. After the burial period had passed, the vessels were excavated and a foodcrust sample was taken from each vessel. All samples were stored at −20°C until protein extraction.

## Protein extraction

Foodcrust protein analysis was performed according to a variant of the SP3 protocol [46] developed by Dr. Virginia Harvey and Dr. Jessica Hendy [47]. In short, 150 µL 6 M guanidine hydrochloride (GuHCl) was added to 20 mg foodcrust, vortexed, centrifuged and incubated at 65°C for one hour. Subsequently, the samples were centrifuged and the foodcrust pellets were discarded. 15 µL of a 100 mM TCEP (Tris(2-carboxyethyl)phosphine) and CAA (2-Chloroacetamide) solution was added to the supernatant, homogenised, centrifuged and incubated at 99°C for 10 minutes. After the incubation, the samples were allowed to cool in room temperature conditions, after which 500 µg of magnetic hydrophilic and hydrophobic beads (Cytiva) and 175 µL of 100% (v/v) ethanol were added to each sample. The samples were then incubated again at 24°C, 1000 RPM for 5 minutes. Protein filtration using the beads was performed by placing the samples on a magnetic rack and allowing the beads to migrate for 2 minutes. The liquid in the samples was then removed until only the beads remained. Samples were taken off the rack and were rinsed with 80% ethanol. This process was repeated three times, the first time 500 µL ethanol was used for the rinsing, the second time 300 µL and the third rinse was done with 200 µL. After the final rinse, the beads were resuspended in 100 µL 50 mM ammonium bicarbonate (ABC). This solution was incubated at 37°C with 750 RPM for 3 minutes, after which the samples were placed back on the magnetic rack. In-solution digestion was performed by adding 1 µL of a 0.2 µg/µL trypsin solution to the samples and incubating them at 37°C, 750 RPM for 18 hours.

After digestion, the samples were centrifuged and placed back on the magnetic rack. Beads and suspended proteins were separated by transferring the solution to another set of Eppendorf tubes. 10 µL of 5% (v/v) trifluoroacetic acid (TFA) was added to the samples to halt digestion. As a last step, the peptides were desalted with C18 ZipTips. The ZipTips were primed twice with a 100 µL 50% acetonitrile (ACN) and 0.1% TFA solution and washed twice with 0.1% TFA. Then the peptides were aspirated from the sample and the ZipTips were washed again twice with 0.1% TFA before eluting the peptides in 50% ACN and 0.1% TFA.

## Protein mass spectrometry analysis

The eluted peptides were analysed using an mClass nanoflow UPLC (Ultra Performance Liquid Chromatography) system (Waters) coupled to an Orbitrap Fusion Tribrid mass spectrometer (Thermo Scientific) following the same protocol as published elsewhere [48]. In short, the UPLC was equipped with a nanoEaze M/Z Symmetry 100 Å C18, 5 µm trap column (180 µm x 20 mm, Waters) and a PepMap, 2 µm, 100 Å, C18 EasyNano nanocapillary column (75 µm x 500 mm, Thermo Scientific). Chromatographic separation of the peptides was performed by first washing the peptides from the trap column onto the capillary column with 0.05% trifluoroacetic acid aqueous solvent at a flow rate of 15 µL/min. After five minutes the flow was switched to the capillary column, which was first washed for seven minutes with solvent A (0.1% FA (formic acid)) and 3–10% solvent B (0.1% FA in acetonitrile). The concentration of solution B in the mixture was steadily increased over time. After these initial seven minutes, the concentration of solvent B was increased to 35% over 30 minutes and lastly to 99% over five minutes. Finally, the column was washed with 99% solvent B for four minutes.

As the peptides were eluted off the column they were introduced into the mass spectrometer, which was equipped with an EasyNano ionisation source (Thermo Scientific). Xcalibur (version 4.0, Thermo Scientific) was used to acquire ESI-MS and MS2 spectra both in positive mode. The following instrument source settings were used: ion spray voltage, 1,900 V;

sweep gas, 0 Arb; ion transfer tube temperature; 275°C. MS1 spectra were acquired in the Orbitrap with: 120,000 resolution, scan range: m/z 375−1,500; AGC target, 4e5; max fill time, 100 ms. Acquisition was data dependent (DDA) and performed in topN mode selecting the 12 most intense precursors with charge states >1. Easy-IC was used for internal calibration. Dynamic exclusion was performed for 50 s post precursor selection and a minimum threshold for fragmentation was set at 5e3. MS2 spectra were acquired in the Orbitrap with: 30,000 resolution, max fill time, 100 ms., HCD; activation energy: 32 NCE. The mass spectrometry proteomics data have been deposited to the ProteomeXchange Consortium via the PRIDE 64 partner repository with the dataset identifier PXD059930 and 10.6019/PXD059930.

## Protein data analysis

The resulting raw files were analysed with MetaMorpheus [49] using the following settings: files were calibrated with a product mass tolerance of 25 ppm and a precursor tolerance of 15 ppm. A G-PTM-D (Global-Post Translational Modification Discovery) task was performed in order to identify the most common post-translational modifications in the samples. The following PTM groups were included in the G-PTM-D task: common biological (34 PTMs), common artefact (10 PTMs), metal (16 PTMs) and trypsin digested (2 PTMs). The precise PTMs that were included can be found in SI 3. To quantify PTM presence occupancy, that is the fraction of a protein that displays a given PTM at a specific position in the protein sequence [50], was calculated automatically at protein level by MetaMorpheus. For the search task a product mass tolerance of 0.5 Da and precursor mass tolerance of 3 ppm was maintained. The maximum number of missed cleavages was 2 and the minimum peptide length was set to 7. Only fully tryptic peptides were accepted.

Using these settings the samples were searched against a custom database consisting of reference proteomes of *Beta vulgaris, Chenopodium quinoa, Spinacia oleracea, Salmo salar* and *Bos taurus*, which were downloaded from the Uniprot protein database [34]. In addition to the proteomes of the species used in the cooking experiments, it was also decided to include quinoa (*Chenopodium quinoa*) and spinach (*Spinacia oleracea*), because of the limited coverage of the *Beta vulgaris* reference proteome. We feared that our analysis would bias against *Beta vulgaris* and therefore decided to include the two species with the largest reference proteomes from the Amaranthaceae family, which includes *Beta vulgaris*. All five proteomes mentioned above were downloaded from UniProt on 03-03-2023 [34]. Lastly, we included MetaMorpheus' inbuilt database of contaminants, cRAP [51], in our search. The exact task files and protein database used in this analysis are included in SI 3.

The MetaMorpheus protein identifications were then filtered following the recommendations for palaeoproteomic identifications [52], rejecting all proteins which were present in the blanks or which were represented by less than two unique peptides.

Statistical analysis was performed in R *v4.1.1* [53] using the Rstudio environment *v2024.4.0.735* [54].

## Foodcrust quantification

In order to investigate the influence of ingredient composition on the amount of foodcrust generated after cooking we quantified the volume of foodcrust using 3D scanning. A 3D scan of each ceramic vessel was acquired before the first cooking event and after the ninth using an Artec Space Spider (Artec 3D). 3D model construction and further analysis of the 3D models were all performed in Artec Studio 16 Professional v16.0.8.2. Volume measurements were obtained by combining the scans for each vessel and time point using the 'Sharp fusion' function. Then the volume of the entire 3D model was calculated using the in-built volume calculation function. The foodcrust volume was calculated by subtracting the before-cooking 3D model volume from the after-cooking 3D model volume.

Vessels 1, 7, 11 and 21 were damaged during the cooking process and some fragments had detached from the ceramic wall. To prevent the loss of ceramic fragments from influencing the foodcrust quantification the ceramic wall from which the fragment had detached were deleted from both the before and after model. The resulting hole in the 3D models was closed using the 'Fix holes' function. Irregularities in the 3D model resulting from this intervention were resolved using

the 'Small-object filter' and 'Hole filling' 'Postprocessing' functions. Foodcrust volume calculations could then be obtained in the same way as for the undamaged experiments.

## Results

A total list of all proteins recovered from the samples can be found in SI 4. There appeared to be substantial differences in the number of proteins that could be identified between the different experimental conditions and the number of cooking events for a particular ingredient combination (Fig 3). Additionally, most samples show large differences in the contribution of the different ingredients to the total amount of recovered proteins. Some samples, 6.5, 7.1, 12.1, 12.9, 15.1 and 21.1, also yielded proteins from a species not included in their mixture (Table 1).

Fig 3 also shows that for most experimental conditions the number of protein identifications decreases as the number of cooking events increases, which fits with the expected increase in protein damage due to extended heating. To further explore this pattern we plotted the distribution of protein identifications per sample point (Fig 4). For each sample we also calculated the protein input in grams based on the ingredients' protein content [55]. As expected the number of proteins identified decreases after the first sampling point and is lowest after the burial, but some of the samples obtained after the ninth cooking event contain more proteins than any sample taken after the fifth event. To test if these differences are statistically significant, a Kruskal Wallis test [56] was performed (p = 4.183e$^{-8}$), which indeed suggests that there are significant differences between the number of proteins recovered after the different cooking events. Post hoc analysis using Dunn's test [57,58] shows that there are significant differences between cooking events 1 and 5 (p = 5.46*10$^{-5}$), 1 and 9 (p = 1.08*10$^{-3}$), 1 and PE (5.78*10$^{-7}$) and lastly between events 9 and PE (p = 0.0497). The seeming difference in protein recovery between events 5 and 9 is therefore not statistically significant.

To further investigate the relationship between the observed proteome abundance and the cooked ingredients we processed the identified peptides in the following way: first, we filtered the identified peptides to only include those that were observed in the single-species samples (i.e., experiment numbers 1, 6 and 21). As the number of identified peptides, let alone shared-peptides, decreases rapidly with increased cooking, we only included samples from the first cooking event. For each protein in each sample, we then selected the three peptides with the highest intensities (Fig 5) and summed their intensities per sample, giving a summed intensity as a proxy of their protein abundance. Using these summed intensities as the response variable we created a linear regression model with the nutritional composition as predictors. In the first model we split the fat content of each sample by saturated, mono-unsaturated and poly-unsaturated fatty acids, while in the second model, we grouped these three fatty acid subcategories together. Both models showed a significant relationship ($R^2$ = 0.7765, F(2,18) = 35.75), the total protein (p = 1.16e$^{-6}$) and carbohydrate (p = 4.84e$^{-6}$) content, in particular, showed a significant positive correlation with the observed sum of intensities.

The same approach also allowed us to identify the proteins that were most abundant across our samples after the first cooking event (Table 2). Of the ten most intense proteins, eight belong to *Salmo salar* and the remaining two are from *Bos taurus*. No plant proteins feature among the most intense proteins. Interestingly, L-lactate dehydrogenase, number nine in the top ten, was only identified in a single sample (vessel 6), which was filled with 100% salmon. Additionally, *Salmo salar* collagen alpha-1(I) is listed twice in the top ten with two different accession numbers. An alignment of the complete sequences of these two proteins via Clustal [59] shows a 93.68% similarity in sequence between the two, indicating that there are a number of differences in the sequences. The UniProt entries of the two proteins indicate that they are both automatically predicted from genomic data, albeit from different gene identifiers [34](accessed on 08-07-2025).

The quantification strategy described above is limited by the severe cooking- and burial-induced degradation of the proteins. The number of proteins that survive all cooking events and the burial is limited across all conditions of mixed foods (Fig 3). Out of the total 623 proteins that were identified across all the cooking experiments, 13 are present in samples from both the first, fifth, ninth cooking event and post-burial (Table 3). Remarkably, of these 13 persistent proteins, three belong to plant species, while none of the *B. taurus* proteins were found in samples from each cooking event. An overview

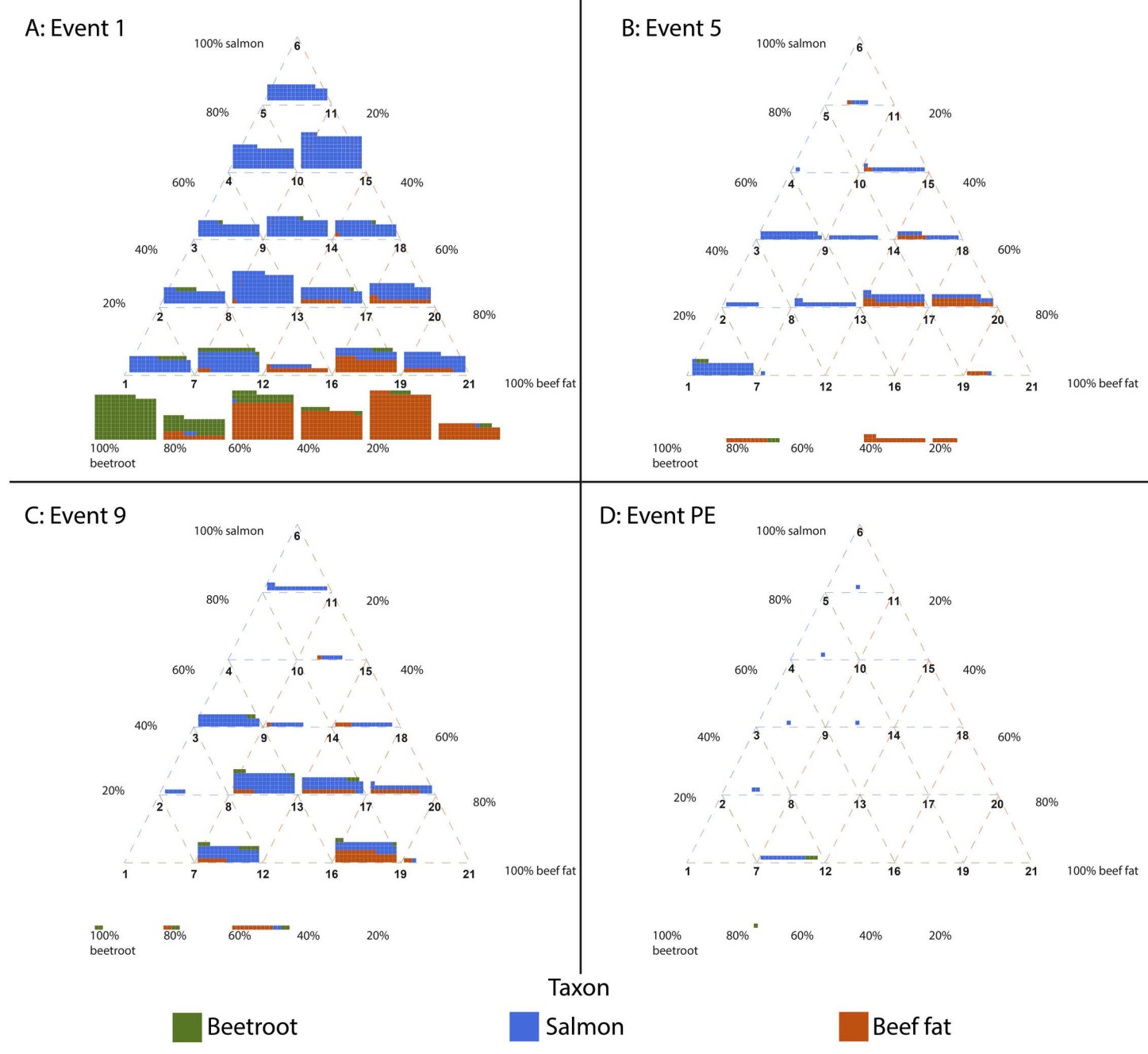

**Fig 3. Number of proteins identified in each experimental condition split up by number of cooking events and taxon.** PE indicates the post-excavation sample. *Beta vulgaris*, *Chenopodium quinoa* and *Spinacia oleracea* are grouped together as 'plant proteins'. Each square represents an identified protein.

of all identified proteins and the number of samples they were recovered in per cooking event (SI 5) shows that there are three additional salmon proteins that were recovered from the post-excavation samples but were not observed in samples from earlier time points. Two, isocitrate dehydrogenase (B5DGS2) and aspartate aminotransferase (B5X142), were not

**Table 1. Identified proteins belonging to taxa not included in the sample's mixture.**

| Experiment number | % Beetroot | % Salmon | % Beef fat | Cooking event | Protein Accession | Protein Full Name | Sequence Coverage | Number of Unique Peptides | Organism |
|---|---|---|---|---|---|---|---|---|---|
| 6 | 0 | 100 | 0 | 5 | A0A3Q1LL35 | Uncharacterized protein | 25% | 2 | Bos taurus |
| 7 | 80 | 20 | 0 | 1 | A0A1S2X522 | phosphopyruvate hydratase | 63% | 4 | Salmo salar |
| | | | | | B5DG55 | Alpha-1,4 glucan phosphorylase | 48% | 5 | Salmo salar |
| | | | | | Q3ZLR1 | Superoxide dismutase [Cu-Zn] | 23% | 2 | Salmo salar |
| 12 | 60 | 0 | 40 | 1 | C0H9B8 | Trifunctional enzyme subunit beta, mitochondrial | 9% | 2 | Salmo salar |
| | | | | 9 | A0A1S3Q7E3 | collagen alpha-1(I) chain-like | 10% | 2 | Salmo salar |
| | | | | | A0A1S3SM10 | collagen alpha-1(I) chain-like | 8% | 2 | Salmo salar |
| 15 | 60 | 40 | 0 | 1 | Q9LEE0 | phosphopyruvate hydratase | 15% | 2 | Spinacia oleracea |
| 21 | 0 | 0 | 100 | 1 | A0A0J7YLM1 | Elongation factor Tu (Fragment) | 13% | 4 | Beta vulgaris subsp. vulgaris |
| | | | | | A0A0J8B1E8 | 50S ribosomal protein L5, chloroplastic | 14% | 2 | Beta vulgaris subsp. vulgaris |
| | | | | | A0A0K9Q7T6 | S1 motif domain-containing protein | 4% | 2 | Spinacia oleracea |

observed in the sample taken after the fifth cooking event, while myosin-7B (A0A1S3QIK3) was not detected after the ninth cooking event. Additionally, some proteins appear to occur consistently and frequently throughout the experiment until the burial, after which they disappear. A notable example is two collagen alpha-1(I)-like entries, A0A1S3Q7E3 and A0A1S3SM10, which are found in 9–11 samples after the first, fifth and ninth cooking event, but in none after burial.

The above results highlight the challenges in linking the abundance of the identified proteins with the species used in the cooking, which was one of the prime aims of this study. Especially, as of the most intense proteins after the first cooking event (Table 2), only alpha-1,4 glucan phosphorylase (B5DG55) and myosin regulatory light chain 2 (Q7ZZN0) could be recovered in samples from all cooking events, highlighting that there are substantial differences in the impact of degradation on particular proteins. Due to the differences in chemical properties between proteins, such variation is not wholly unexpected. Recently, hydrophobicity (refs) has been gaining attention as a potentially important factor in protein survival. Therefore, we calculated the hydrophobicity of the observed fraction of our identified proteins (Fig 6) using the whole residue values as determined by Wimley and White [60].

It is clear that the overwhelming majority of identified proteins are hydrophilic, or at least the part of the protein sequence that we identify is hydrophilic. This applies to all samples and across all cooking events, even though recent studies suggest that hydrophobicity may aid protein survival [30, 61]. Consequently, the strong bias towards hydrophobic parts of the protein sequences may be more indicative of biases in the protein extraction process rather than resistance against degradation during periods of heating and burial.

Apart from degrading proteins to the point they can no longer be detected, it is well known that processes such as heating can lead to PTMs forming, although the precise effects are not well understood [62, 63]. Our G-PTM-D search identified 43 different PTMs (SI 5), some of which were near ubiquitous, while others were extremely rare. The PTM profile of the identified PTMs on the most intense proteins from the first cooking event seems fairly consistent across all cooking

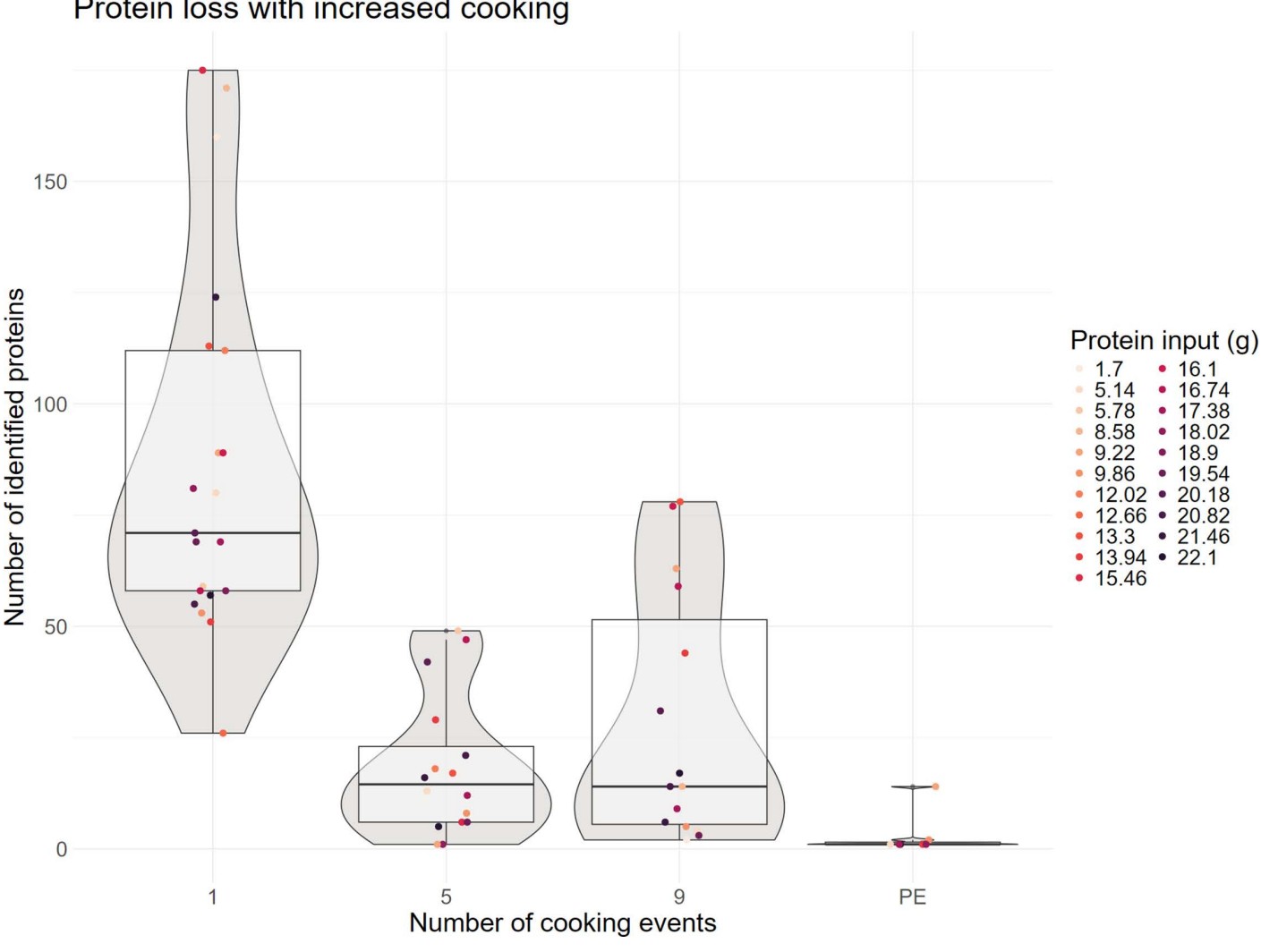

**Fig 4. Distribution of the number of proteins identified per sampling point.** In general, the number of identified proteins decreases with increasing cooking events, with two vessels acting as outliers.

events (Fig 7). Some changes in the average occupancy of PTMs can be observed, such as the increase in formylation on K and deamidation in hemoglobin subunit beta

(P02070) and beta-enolase (A0A1S2X522), yet simultaneously N deamidation seems to decrease in L-lactate dehydrogenase (A0A1S3MJX7) and one of the salmon collagen alpha-1(I) chain-like proteins (A0A1S3SM10). Similarly, phosphorylation on S and T seems to increase in bovine collagen alpha-1(I) chain P02453, but decreases in the other salmon collagen alpha-1(I) chain-like protein (A0A1S3Q7E3).

To test if there was a clearer pattern in PTM occupancy in the larger dataset, we tested for differences between the cooking events and the occupancy using the Kruskal Wallis test for each of the 43 identified PTMs. In case the Kruskal Wallis test indicated a significant difference ($p < 0.05$), a post-hoc Dunn's test was performed to identify between what cooking events the occupancy differed. Statistically significant differences were found between:

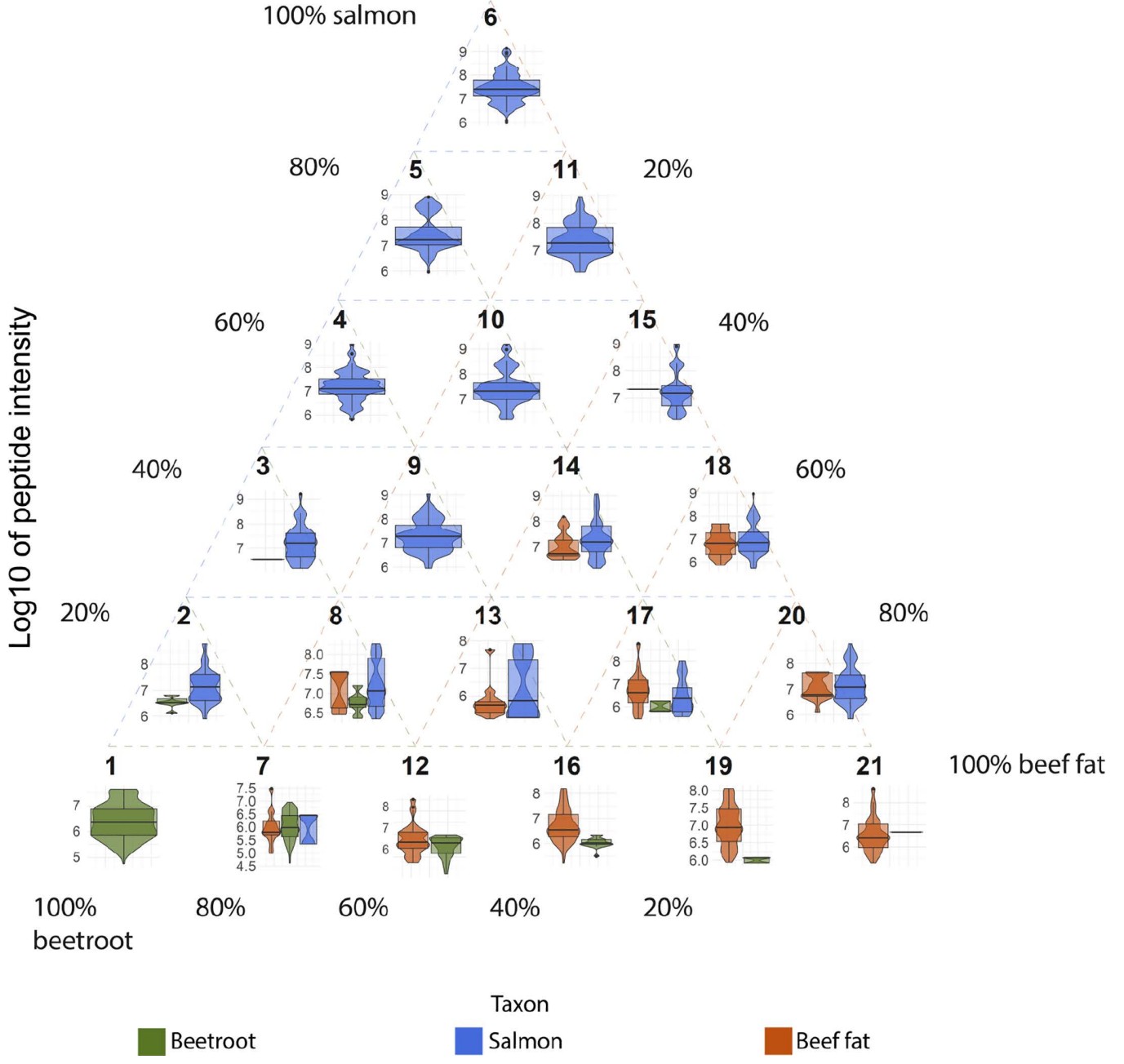

**Fig 5. Distribution of the intensity of the three most intense peptides per protein.** The y-axis shows the log 10 of intensity. For the ingredient combination of each experiment number see Fig 1.

- Hydroxylation on P ($\chi^2 = 23.72$, $p = 2.858e^{-5}$), significant differences ($p < 0.05$) between all events, except between cooking event 5 and 9.

- Deamidation on N ($\chi^2 = 28.582$, $p = 2.742e^{-6}$), significant differences between cooking events 1 and 5 and 1 and 9.

- Sodium on E ($\chi^2 = 10.612$, $p = 0.01402$), only differs significantly between cooking events 1 and 5.

**Table 2. Top 10 most intense proteins summed across all samples from cooking event 1.**

| Protein accession | Protein full name | Taxon | Average sequence coverage (%) | Number of samples protein was present in | Summed intensity |
|---|---|---|---|---|---|
| B9EP57 | Troponin C, skeletal muscle | Salmo salar | 61% | 11 | 1.47E+10 |
| A0A1S2X522 | Phosphopyruvate hydratase | Salmo salar | 77% | 15 | 9.99E+09 |
| P02453 | Collagen alpha-1(I) chain | Bos taurus | 35% | 6 | 5.31E+09 |
| Q7ZZN0 | Myosin regulatory light chain 2 | Salmo salar | 84% | 15 | 4.64E+09 |
| B5DG55 | Alpha-1,4 glucan phosphorylase | Salmo salar | 57% | 16 | 4.34E+09 |
| B5DGL9 | Fructose-bisphosphatase | Salmo salar | 48% | 12 | 3.37E+09 |
| A0A1S3Q7E3 | Collagen alpha-1(I) chain-like | Salmo salar | 34% | 10 | 2.72E+09 |
| P02070 | Hemoglobin subunit beta | Bos taurus | 80% | 9 | 2.65E+09 |
| A0A1S3MJX7 | L-lactate dehydrogenase | Salmo salar | 82% | 1 | 1.88E+09 |
| A0A1S3SM10 | Collagen alpha-1(I) chain-like | Salmo salar | 24% | 11 | 1.41E+09 |

**Table 3. Uniprot accession number, full name and associated species of the proteins found in at least one rim sample from each sampling point. The values indicate the number of samples each protein was identified in per sampling point.**

| Protein Accession | Protein Full Name | Organism | 1 | 5 | 9 | PE |
|---|---|---|---|---|---|---|
| B5DG55 | Alpha-1,4 glucan phosphorylase | *Salmo salar* | 16 | 10 | 11 | 4 |
| A0A0J8E469 | rRNA N-glycosylase | *Beta vulgaris* subsp. *vulgaris* | 12 | 2 | 8 | 2 |
| B5DFU7 | AMP deaminase | *Salmo salar* | 10 | 6 | 7 | 2 |
| A0A1S3PV75 | Glucose-6-phosphate isomerase | *Salmo salar* | 12 | 8 | 10 | 1 |
| A0A1S3MVJ3 | ATP synthase subunit beta | *Salmo salar* | 14 | 5 | 7 | 1 |
| Q7ZZN0 | Myosin regulatory light chain 2 | *Salmo salar* | 15 | 4 | 7 | 1 |
| B5X4K4 | L-lactate dehydrogenase | *Salmo salar* | 14 | 8 | 6 | 1 |
| A0A1S3KNQ3 | 2-iminobutanoate/2-iminopropanoate deaminase | *Salmo salar* | 5 | 5 | 6 | 1 |
| B5DFX8 | Phosphoglycerate kinase | *Salmo salar* | 6 | 3 | 4 | 1 |
| B5DGZ1 | ADP/ATP translocase | *Salmo salar* | 10 | 2 | 4 | 1 |
| A0A0J8B2U8 | Uncharacterized protein | *Beta vulgaris* subsp. *vulgaris* | 9 | 1 | 3 | 1 |
| A0A803LHN7 | Alcohol dehydrogenase | *Chenopodium quinoa* | 7 | 1 | 3 | 1 |
| B5XDB2 | D-dopachrome decarboxylase | *Salmo salar* | 7 | 2 | 2 | 1 |

- Potassium on E ($\chi^2 = 6.4568$, $p = 0.03962$), significantly decreases between cooking events 1 and 9.

- Acetylation on K ($\chi^2 = 10.939$, $p = 0.004214$), significant difference between cooking events 5 and 9.

Although these five PTMs are statistically significantly different between cooking events, the lack of data for some cooking events for in particular acetylation on K, potassium on E and to a lesser extent sodium on E raise some questions regarding the robustness of these findings (Fig 8). Similarly, hydroxylation on P seems to consistently display a high occupancy, but was hardly detected in the samples from after the burial.

Of secondary importance was to identify the tissue types based on the identified proteins. One theoretical advantage of proteins over other biomolecules is that they may reflect particular tissue-types. In order to achieve this we used STRING (version 12.0) [64,65] to create protein networks for each of the three species and downloaded the tissue expression analysis, which compares the genes associated with the identified proteins to a database (TISSUE) of gene expression profiles per tissue type [66]. The output of this analysis is a list of tissue types, gene names and various metrics regarding the robustness of the tissue association. We filtered this list of tissue associations to only include tissues with a false discovery rate below 0.01 and a minimum number of 35 supporting proteins. The remaining tissue-gene associations were

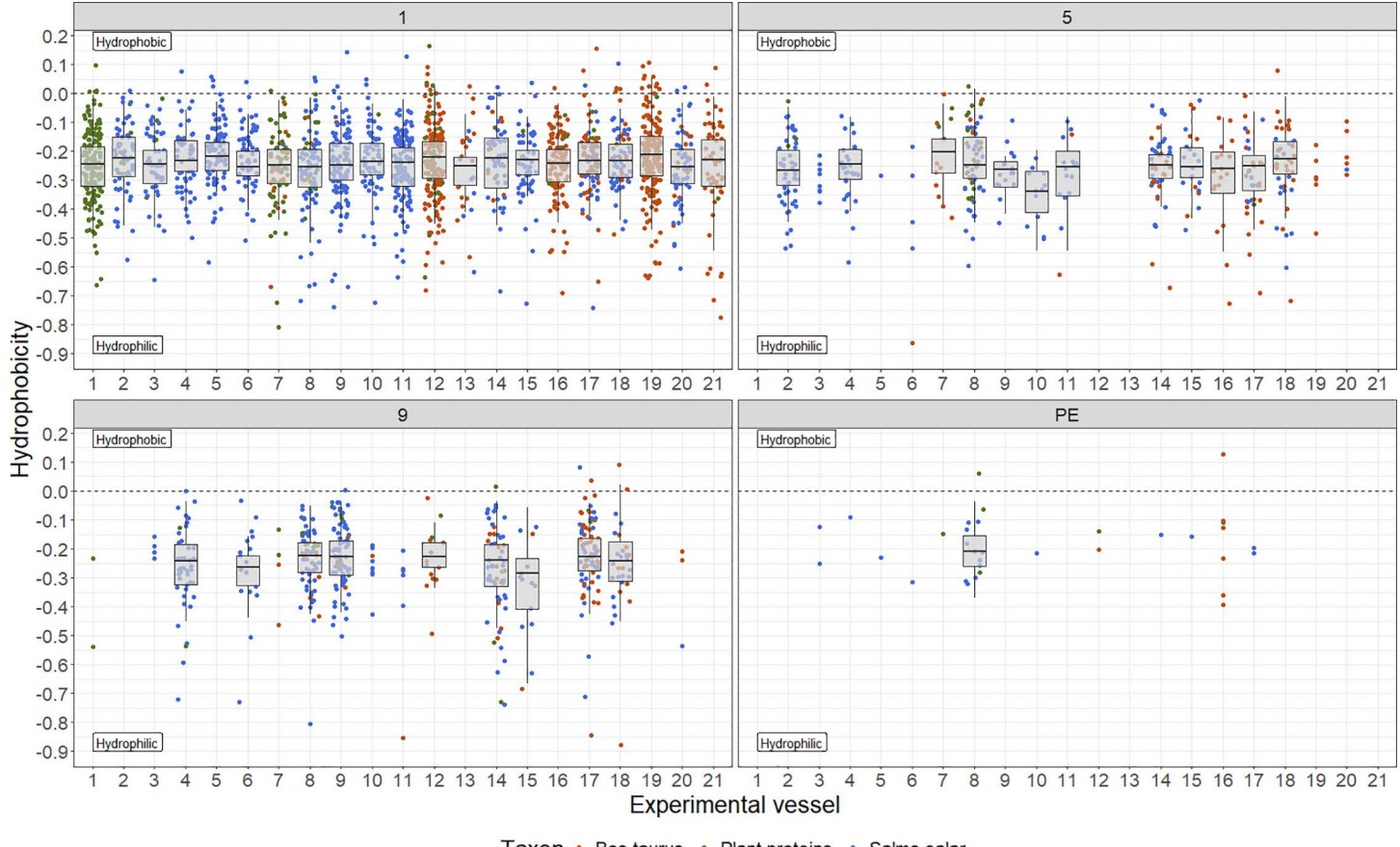

**Fig 6. Hydrophobicity of the identified fraction of each protein for each experimental vessel split by cooking event.** Each dot represents a single protein and is coloured by its taxonomic origin. Hydrophobicity was calculated using amino acid hydrophobicity values following Wimley and White [60].

linked back to the list of proteins identified per sample (Fig 9). Unfortunately, the TISSUE database did not include any data for the observed plant proteins, so it was not possible to associate any particular tissues with the plant proteins.

The tissue associations from detected salmon and beef proteins are mostly unspecific, e.g., "animal" or "whole body". Several associations with heart-related tissues, such as the heart itself, but also the right atrium and ventricles, were observed. One could argue to group these together as one large cluster associated with the cardiovascular system. We chose to keep them separate, as it highlights a disadvantage of this kind of tissue annotation, namely the unspecificity of the observed proteins. The fact that the average number rounded down of associated tissues per gene in our dataset is 5±2 illustrates well how generic most of these genes are. This also explains why tissues such as the liver and various glands are listed in Fig 6 as well. Despite the lack of a clear muscle-associated proteome, Fig 9 highlights the difficulty in distinguishing between raw fats and muscle based on gene expression annotations, as well as showcases the breadth of tissue associations that can be expected from a foodcrust derived from these tissues.

Foodcrust accumulation showed sizeable variation between samples (Fig 10). A linear regression model of the percentage of the ingredients and the foodcrust volume yielded no statistically significant relationships and had an adjusted $R^2$ value of 0.16, showing that the relative abundance of the three ingredients could only explain a small portion of the foodcrust volume variability. A linear regression model of the nutritional content showed a similar pattern. The overall model had a $R^2$ value of 0.16 and neither the absolute input of proteins, carbohydrates or fats yielded a significant relationship.

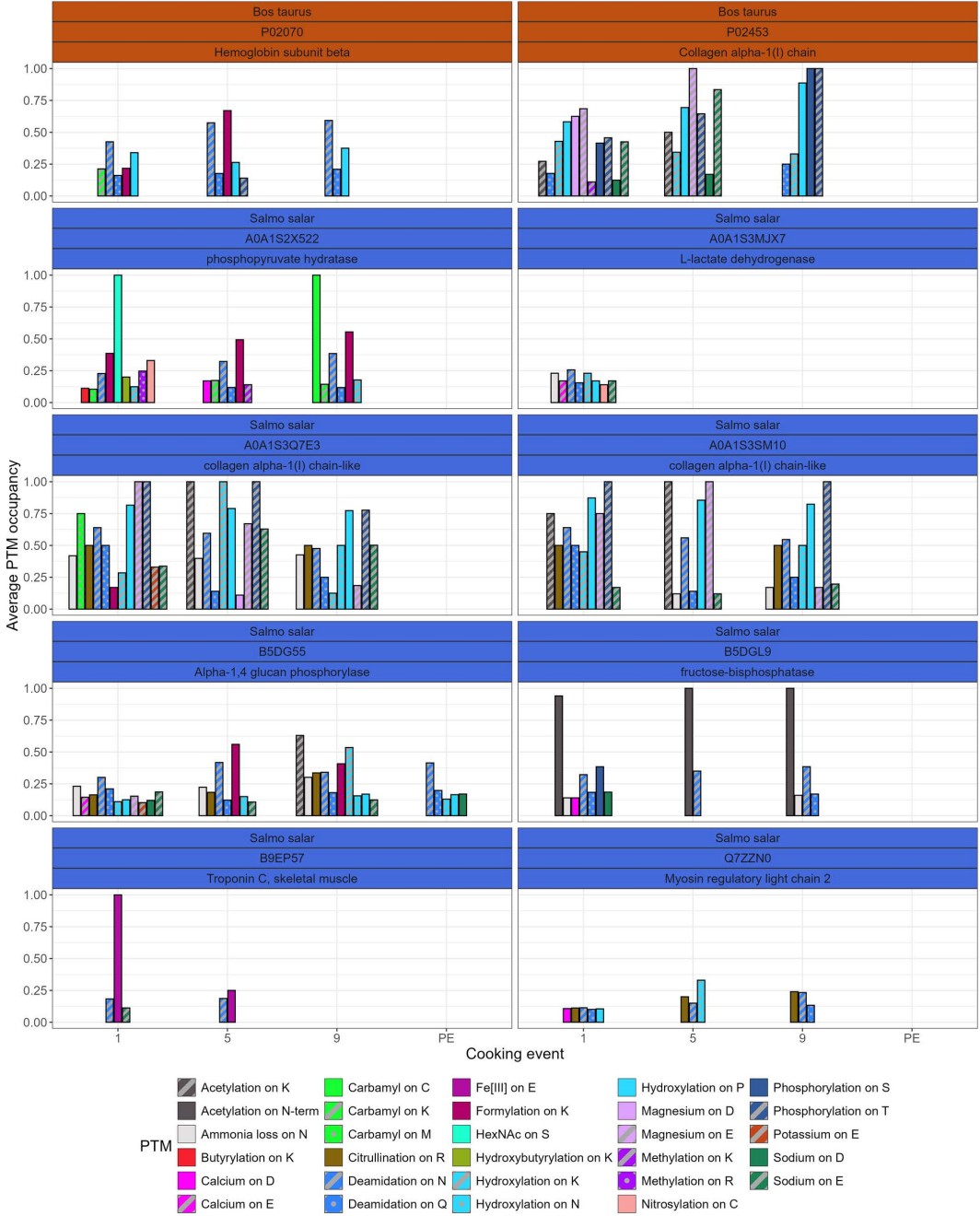

**Fig 7. Average occupancy of PTMs across all cooking events on the 10 proteins that were most intense after the first cooking event.** Only PTMs that had a minimum average occupancy of 10% were included.

## Discussion

### Protein recovery

Our results show large variations in both the number of recovered proteins as well as the relative taxonomic composition of the recovered proteome between the different samples. However, the only significant relationship that we could

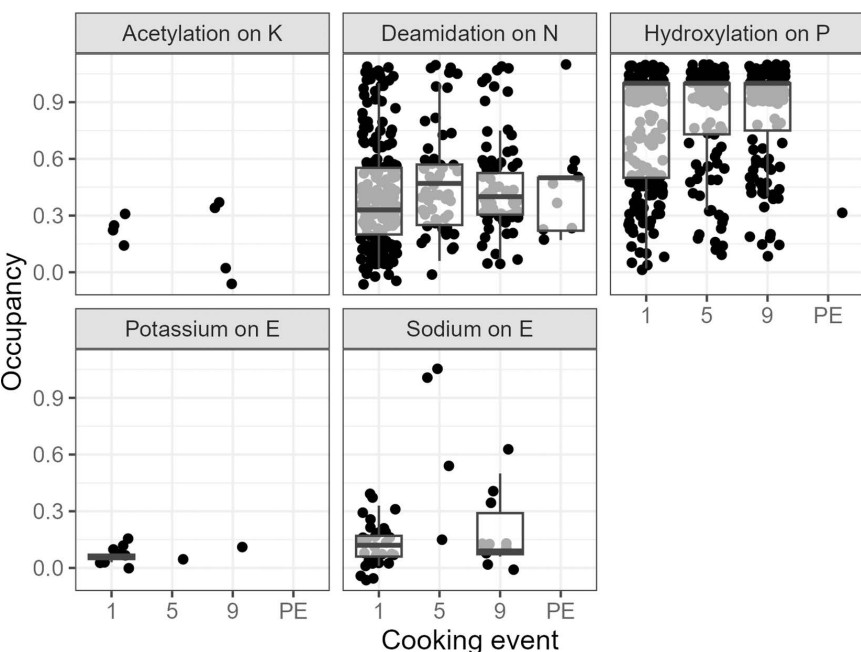

**Fig 8. Distribution of the occupancy of the five PTMs statistically significantly different between cooking events.** Boxplots show the second and third quartile, as well as the median of the distribution.

observe was a decrease in the number of proteins after more cooking events. Such a decrease in protein recovery is not unexpected as it is well known that proteins are vulnerable to high temperatures [67,68] and a decrease in protein recovery from fresh foodstuffs to foodcrust has been observed in previous studies [30]. In fact, one of the aims of cooking is to make food more easily digestible by opening up closed protein structures [63, 69]. Unfortunately, denatured proteins are more vulnerable to degradation [70,71], which bodes ill for protein extraction from archaeological vessels. Although the decrease in the number of recovered proteins with increased cooking events demonstrates the negative effects of prolonged heating, the generally low number of proteins found after the first cooking event indicates that even limited exposure to heat can already induce severe protein degradation.

Heating is also known to induce additional PTMs, yet this pattern was not clearly replicated in our data. Five PTMs showed statistically significant differences between the cooking events, but the limited number of observations for sodium on E, potassium on E and acetylation on K call into question the meaningfulness of these statistical relationships. In contrast, the increasing trend in occupancy of N deamidation with more advanced cooking events is supported by a fairly sizeable set of observations. Deamidation has been linked to protein degradation in archaeological samples [72–74] and an increase in deamidation after increasing cooking duration and burial is in line with our expectations. Although we could observe no statistically significant change in the Q deamidation.

However, as this study does not include unheated samples, it may be that the main effect of heating on PTMs occurs during initial heating rather than prolonged heating after partial charring has already taken place. Instead, the key message of Fig 6 is rather the diversity and high occupancy of many of the PTMs. Open searches, like a G-PTM-D, are not standard practice in palaeoproteomics, and in most studies a select number of PTMs, such as deamidation (NQ) and oxidation (M) are included. Our results show that foodcrusts contain many more PTMs and by calculating the occupancy rather than the percentage of modified residues we can show that the occurrence of PTMs at particular places in the

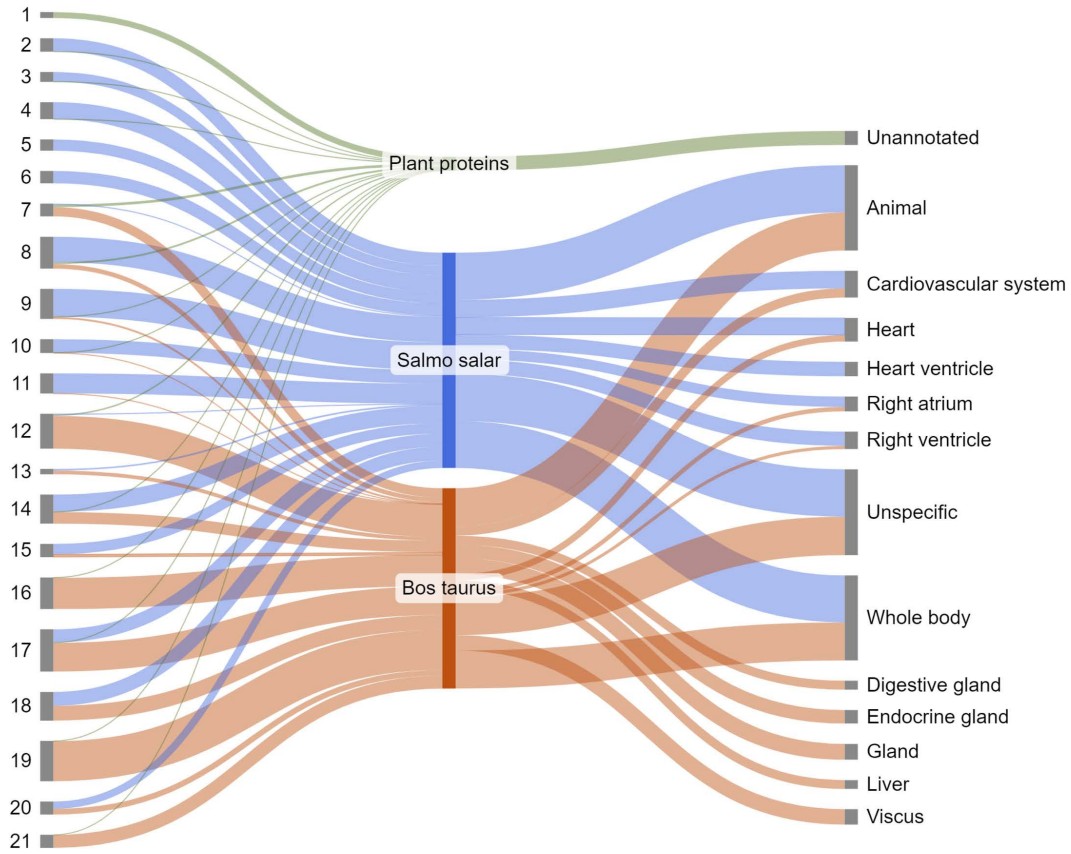

**Fig 9. Tissue associations of the identified proteins per species.**

protein sequence is consistent. Although our results do not allow us to conclude that these PTMs are the result of heating, protein identification from foodcrusts may be more limited if a large number of PTMs are not included.

Apart from the heating-induced damage, biases in the incorporation of proteins in the foodcrusts and extraction efficiency may have contributed to the limited size of the extracted proteome. For example, Fig 6 shows that in the case of nearly all recovered proteins, the part of the sequence that we are able to recover is hydrophobic. Additionally, the fact that the standard deviation is roughly equal to the mean number of detected proteins for cooking events 5, 9 and post-excavation, plus the large variation in how different experimental conditions respond to increased cooking in our opinion suggests that there might be considerable variation in protein preservation across the foodcrust in one pot. The degree of spatial variation in foodcrust protein content might best be tested with a spectroscopy-based method, such as NIR [75]. A better understanding of the amount of spatial variation as well as the scale at which it occurs could be of great benefit in maximising the proteomic data obtained from these degraded samples.

Furthermore, recent studies highlight that the current state of proteomic bioinformatic analysis only manages to assign a protein identification to a fraction of the extracted compounds [76,77]. It may be that future proteomic bioinformatic developments will significantly improve our ability to identify proteins in foodcrust samples, yet for now, we were only able to identify a small set of proteins, which for some samples do not represent all the taxa used in the cooking mixture.

All these issues may have impacted our ability to extract and identify proteins from our experimental samples, but ancient foodcrusts may have been exposed to higher temperatures, for longer periods of time and more often.

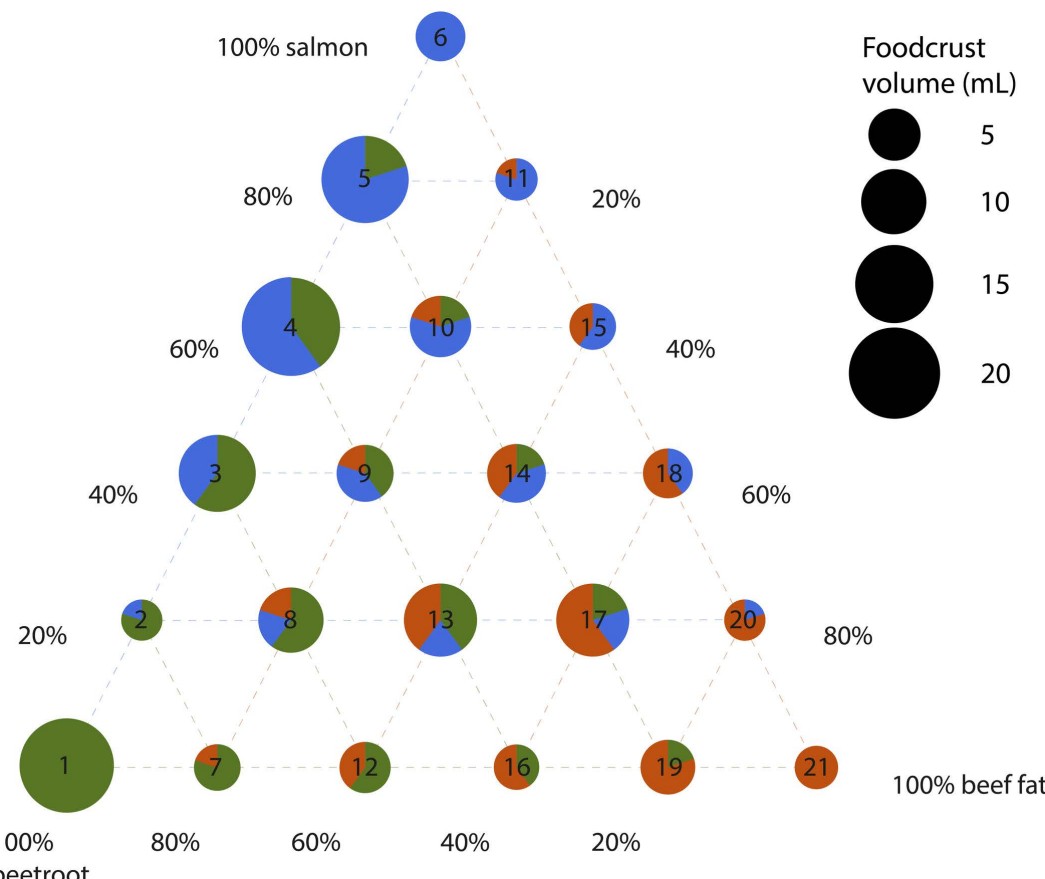

**Fig 10. Volume of foodcrust accumulated after nine cooking events, sorted by experimental conditions listed in** Fig 1**.** The input ingredient composition of each of the experiments is shown by the pie chart on top of the bars. Increasing the proportion of one ingredient over another does not seem to have influenced foodcrust formation.

Nevertheless, previous studies have shown that it is possible to extract proteins from archaeological foodcrusts, but the number of recovered proteins is often in the single digits [4,29]. Considering the limitations of recovering proteins from foodcrusts, it is important to evaluate how well we can succeed in using the limited number of found proteins to identify the taxa and tissues used during the cooking process. Especially in comparison to lipid analysis, which is the most common method for analysing pottery use [78].

### Disentangling mixtures & taxonomic representativeness

One of the main theoretical advantages of proteomics over lipid analysis is the ability to distinguish the different resources in a mixture. As interpretations of resource use in lipid analysis are often based on compound-specific isotope ratios, samples with values that do not fall in the established range of the references are more challenging to interpret. Lipid bio-markers could help with this problem, but they are relatively limited, both in terms of the number of taxa with established biomarkers as well as the taxonomic specificity of the biomarkers [19,43,79,80]. Alternatively, one can apply a mixing model to calculate what combination of foodstuffs could explain the observed values. Mixing models are known to work well in estimating the proportion of a limited number of resources with distinct isotope ratios but struggle more with more

complex mixtures of a larger number of foodstuffs. Additionally, species with very similar ecological niches will be nearly impossible to distinguish isotopically [81].

In theory, proteomic analysis does not suffer from these limitations. Mixing of multiple foodstuffs should not impact our ability to identify them and proteins can allow for up to species-level identifications [82], although this is dependent on the type of recovered proteins and their coverage. Although these theoretical advantages of proteomics might lead to high expectations regarding the performance of proteomic analysis of foodcrust, the aforementioned issues in recovering proteins should temper them. To illustrate this point, 25 out of the 84 samples analysed in this study yielded no proteins at all. 32 samples yielded proteins from all species that were cooked in them. In 18 samples no proteins of one of the cooked species could be found and in 4 samples two species were missing. Lastly, in five samples proteins were found that belonged to a species that was not part of the mixture cooked in that pot (Table 1). The peptides that were uniquely associated with these proteins were run through BLAST [83] to verify their taxonomic identification. The most specific taxon represented by at least two peptides was accepted as the new taxonomic identification (Table 4). This authentication step indicated that the plant proteins in samples 15.1 and 21.1 were shared up to the level of Eukaryota and could thus well derive from the ingredients cooked in the vessels. Although the BLAST taxonomic identifications are more specific for some of the remaining proteins than the others, it does appear that in samples 6.5, 7.1, 12.1 and 12.9 there are proteins present from a species that was not used in the cooking.

There are several explanations for these 'lost' proteins, which occur in 5 out of 84 samples. It could be that the contamination occurred during the cooking experiment or during sampling, although sampling equipment was cleaned with ethanol after each sample. Contamination could have occurred during protein extraction, but all proteins present in the blank were not included in further analysis. Another potential explanation is machine carry over. The fish-specific peptides detected in samples 6.1, 12.1, 12.9 were all also detected in the samples run before them. It may have happened that proteins from a previous sample remained stuck on the column and were only eluted with the next sample. Carry over seems a less likely explanation for the *B. taurus* protein in sample 6.5, as it was the first sample in its batch. Instead, it may be that this protein, A0A3Q1LL35, is background contamination. It is listed as an uncharacterised protein and the

**Table 4. Revised taxonomic identifications using BLAST of potentially intrusive proteins.**

| Experiment number | Cooking event | Protein Accession | Protein Full Name | Sequence Coverage | Number of Unique Peptides | Organism | BLAST taxon |
|---|---|---|---|---|---|---|---|
| 6 | 5 | A0A3Q1LL35 | Uncharacterized protein | 25% | 2 | Bos taurus | Boreoeutheria |
| 7 | 1 | A0A1S2X522 | phosphopyruvate hydratase | 63% | 4 | Salmo salar | Clupeocephala |
| | | B5DG55 | Alpha-1,4 glucan phosphorylase | 48% | 5 | Salmo salar | Salmonidae |
| | | Q3ZLR1 | Superoxide dismutase [Cu-Zn] | 23% | 2 | Salmo salar | Salmo salar |
| 12 | 1 | C0H9B8 | Trifunctional enzyme subunit beta, mitochondrial | 9% | 2 | Salmo salar | Teleostei |
| | 9 | A0A1S3Q7E3 | collagen alpha-1(I) chain-like | 10% | 2 | Salmo salar | Salmoninae |
| | | A0A1S3SM10 | collagen alpha-1(I) chain-like | 8% | 2 | Salmo salar | Salmoninae |
| 15 | 1 | Q9LEE0 | phosphopyruvate hydratase | 15% | 2 | Spinacia oleracea | Eukaryota |
| 21 | 1 | A0A0J7YLM1 | Elongation factor Tu (Fragment) | 13% | 4 | Beta vulgaris subsp. vulgaris | Eukaryota |
| | | A0A0J8B1E8 | 50S ribosomal protein L5, chloroplastic | 14% | 2 | Beta vulgaris subsp. vulgaris | Eukaryota |
| | | A0A0K9Q7T6 | S1 motif domain-containing protein | 4% | 2 | Spinacia oleracea | Eukaryota |

taxon assigned by BLAST, Boreoeutheria, encompasses most of the placental mammals. Consequently, this protein could have derived from any number of common contaminants that could have been introduced at any stage of the experiment.

As with the overall number of proteins, the representativeness of the extracted proteome decreases substantially with additional cooking episodes. After the first cooking event 5 of the 21 samples were missing one of the input foods, but after burying the samples 6 out of 7 samples were missing proteins from one or two of the cooked foods. Interestingly, despite 3 of 7 samples yielding protein after burial containing beef fat, none yielded *Bos taurus* proteins. To check if there was a systematic bias in recovery against any particular species we calculated the recovery rate of each species by dividing the number of samples they were found in by the number of samples that the species was cooked in (Table 5). The aforementioned 6 samples with proteins from species that were not part of the mixture were excluded from the recovery rate calculation for that particular species. The protein abundance of each species was not taken into account in calculating the recovery rate due to the previously discussed challenges in quantifying protein abundance for degraded samples.

Table 5 shows clear differences in recovery rate between the foodstuffs. In general, beetroot was the most challenging to recover, which is in line with expectations considering the low protein content of beetroot. Vice versa, the fact that salmon has the highest recovery rate is in line with it having the highest protein content. Additionally, although in general there is a pattern of lower numbers of proteins and decreased recovery rate with an increased number of cooking events, the recovery rate for beetroot and beef is higher for cooking event 9 than 5. However, the Kruskal Wallis and Dunn's test showed that cooking events 5 and 9 were the only two groups without a statistically significant difference in protein numbers. The lack of a statistically significant difference may be explained by the addition of fresh food at the start of each cooking event. It may be that it is this 'freshest' food incorporated into the foodcrust that is yielding us proteins.

Secondly, Table 5 highlights the sudden disappearance of *Bos taurus* proteins after burial. It is unclear what causes the lack of *Bos taurus* protein preservation. It could be that the proteins in beef fat (which includes vimentin, creatine kinase B-type and heat shock protein beta-1) are simply more vulnerable to degradation due to their structure. Although Fig 5 shows that the *Bos taurus* proteins identified are associated with largely the same tissues as the *Salmo salar* proteins. Alternatively, the vulnerability to degradation may be due to the structure of the foodcrust and how fat-rich foods are incorporated in them rather than the structure of the proteins. There is some reason to suspect other factors might also be at play. A recent study performed a similar experiment, generating and burying foodcrusts made using red deer (*Cervus elaphus*), salmon (*Salmo salar*) and sweet chestnut (*Castanea sativa*) [30]. Although our experiment did not use *Cervus elaphus*, it may be compared to our vessels with *Bos taurus*. The two species are closely enough related that most of their proteins should be similar, if not their sequences. Of their top 5 most abundant deer proteins in the foodcrust after cooking by peptide count, only fructose-bisphosphate aldolase (A0A3S5ZPB0) was detected in our samples and only 2 samples from the first cooking event. Two of the other common deer proteins, troponin T3 and myosin-1 were not directly detected in our samples, but myosin light chain 1/3 (A0JNJ5) and troponin I2 (F6QIC1) were detected, although again in a limited number of samples. As for the most abundant proteins after cooking, only myoglobin was detected in our samples, but bovine myoglobin is the 7th most frequently recovered bovine protein in our samples. However, it was not found in any samples after burial.

**Table 5. Recovery rate of each species per cooking event.**

| Cooking event | Beetroot | Salmon | Beef |
|---|---|---|---|
| 1 | 80% | 93% | 87% |
| 5 | 27% | 93% | 67% |
| 9 | 53% | 80% | 73% |
| PE | 13% | 40% | 0% |
| Average | 43% | 77% | 57% |

There is more overlap in the *Salmo salar* proteins between our study and Evans *et al.* (2024). All five most abundant *Salmo salar* proteins from their after-cooking samples were observed in our samples, although they do not occur in the top 10 of most frequently recovered *Salmo salar* proteins in our samples. As for the most abundant proteins after burial, four out of five were detected in our samples, but only one, myosin heavy chain, fast skeletal muscle-like (A0A1S3QIK3), in one of our after-burial samples.

The recovery of sweet chestnut and beetroot cannot be compared as easily as *Cervus elaphus* and *Bos taurus*, but it is noteworthy that they recover chestnut proteins less frequently than *Salmo salar* or *Cervus elaphus* proteins in their buried samples, whereas we do find plant proteins, but no *Bos taurus* proteins in our buried samples. Of course, their experiment used deer meat, while we used beef fat and it may be that this changes the preservation potential of the proteins contained within the tissue. Regardless of what prevented us from recovering *Bos taurus* proteins, it is clear that, especially for buried samples, the recovered proteome might not reflect all the foodstuffs cooked in the vessel and that there is substantial variability in our ability to obtain proteins from buried samples.

Compared to other proteomic analyses of organic residues, the results presented here differ somewhat in the diversity of proteins identified. Previous cooking experiments (without burial) on protein extraction from ceramics found mostly myosin, collagen and haemoglobin type proteins [84, 85]. Myosins and collagens have also been found in the limited number of studies on archaeological foodcrust [29], as well as parvalbumin and vitellogenin in one specific study [4]. Calcified residues have also yielded myosin and haemoglobin proteins [86], but in general this type of residue seems to more frequently yield dairy proteins [26, 28]. Collagen, myosin and haemoglobin also feature among the most intense proteins we observed after the first cooking event (Table 2), but only myosin persisted through all cooking events and burial (Table 3). Instead, most of the other proteins listed in Table 3 are not commonly reported from organic residues. Data from other types of organic residues, such as from the ceramic matrix or calcified residues are not necessarily directly comparable to foodcrusts. Although these materials also contain dietary proteins, each of them has their own unique properties and processes of protein incorporation. Nonetheless, based on previous studies it seems that for meat-based foodstuffs proteins such as collagens, myosin and haemoglobins form a 'core' proteome that can be relatively frequently recovered.

## Foodcrust abundance

Our and previous results [30] show that the protein contents of foodcrusts do not 'fairly' represent all the foodstuffs prepared in the vessel. Biases in protein recovery for particular foodstuffs were expected to a certain extent, but Fig 10 suggests that there are also differences in the quantity of foodcrust that particular foodstuffs generate. Fig 10 shows that there is not one particular food that is driving the formation of more foodcrust. Overall, the beetroot-salmon mixtures produce the most foodcrust, followed by beetroot-salmon-beef fat mixtures. Beetroot-beef fat or salmon-beef fat seems to produce the least foodcrust, suggesting that the presence of such a fat-rich food is not necessarily a driver of foodcrust formation. Indeed, nowadays in contact frying it is common to add a lipid-rich substance, such as oil or butter, to prevent food from sticking to cookware. This common practice is supported by studies showing that increasing amounts of oil reduces the adhesion of food to containers [87, 88]. Lipid-based coatings have even been proposed for pharmaceutical containers to prevent nonspecific binding of the active components of drugs to container walls [89]. Beyond food content, experimental studies on organic residue formation, or 'fouling' [90], show that the material characteristics (e.g., surface roughness) have an substantial effect on the non-stick properties of cookware, as well as the ease of cleaning [88, 91].

The key finding of Fig 10 is perhaps to highlight the importance of carbohydrate and protein-rich foodstuffs in generating foodcrust in the first place, while fat-rich foods seem to play a lesser role. Yet the most common method of analysing these organic residues has so far been lipid analysis, which may bias our interpretation of the processed resources towards fat-rich foodstuffs. Such a variable pattern in foodcrust formation in our dataset emphasises the value of replicates, which we, unfortunately, are not able to provide. Nonetheless, we argue that these results are a valuable addition to

our current understanding of the formation of organic residues, especially because as far as we are aware there has been no absolute quantification of the abundance of foodcrust formed up to this date.

Additionally, although we could not obtain quantitative measurements of the volume of foodcrust forming on different parts of the pot (e.g., rim, body or bottom), it is clear from Fig 2 and the 3D scans that foodcrusts predominantly, and in the case of some vessels only, at the rim of the vessel. In particular, foodcrusts seem to form mostly at the interface between the water level and the ceramic, which seems to be in line with observations on the formation of 'fouling' in industrial dairy processing contexts [92]. A potential explanation for the proclivity of foodcrust formation at this position might be the steep gradient in temperature. Most of the food during the cooking events was submerged in water, which cannot exceed its boiling point of 100 °C, whereas the ceramic itself will be closer to the 270 °C the experiment was conducted at. As the waterline receded through evaporation more of the food would suddenly be exposed to the higher temperatures, potentially prompting the formation of foodcrust. This applies to both pieces of the solid foodstuffs, as well as any soluble components.

## Conclusion

This experiment had three core aims: firstly, to determine the relationship between the relative ratios of input foods and the proportions between the species of the recovered proteins. Secondly, to measure the impact of repeated cooking and burial on protein preservation and lastly how the different ingredient combinations impacted the amount of foodcrust that formed.

The data shows clearly that the relative abundance of recovered proteins does not mirror the actual input of foodstuffs. Most identified proteins belonged to *Salmo salar,* followed by *Bos taurus* and lastly *Beta vulgaris*. This pattern applied to samples taken after cooking events 1, 5 and 9. However, in the samples taken after the burial no *Bos taurus* proteins could be recovered, although a small number of *Beta vulgaris* proteins were identifiable. These results highlight that not only do food input and recovered proteins not necessarily align, the relative abundance of taxa may change due to increased degradation.

Secondly, the number of recovered proteins decreased significantly from the first to the fifth cooking event, plateaued from the fifth to the ninth and decreased again from the ninth event until after burial.

Lastly, no statistically significant predictor of the foodcrust volume could be identified, but the mixtures that formed the most foodcrust in terms of volume were beetroot-salmon, followed by a mixture of all three ingredients, suggesting that lipid-rich foods are not as important for the volume of foodcrust.

## Supporting information

**S1 Data. Nutritional composition of the used foods.**
(CSV)

**S2 Data. Overview of animal and plant species identified at Danish Ertebølle sites and the availability of public protein reference data for these species.**
(XLSX)

**S3 Data. Metadata of the MetaMorpheus database search.**
(ZIP)

**S4 Data. List of identified proteins.**
(XLSX)

**S5 Data. Overview of the number of the number of foodcrusts per cooking event a protein was identified in.**
(XLSX)

**S6 Data. Overview of the identified PTMs, both per protein and summarised per cooking event.**
(XLSX)

## Acknowledgments

We thank the Bioscience Technology Facility and Chemistry Department at the University of York for mass spectrometry access and support, in particular we would like to thank Dr. Adam Dowle for his support in running the samples and feedback on the manuscript. The instrumentation is part of the York Centre of Excellence in Mass Spectrometry. The centre was created thanks to a major capital investment through Science City York, supported by Yorkshire Forward with funds from the Northern Way Initiative and subsequent support from EPSRC (EP/K039660/1; EP/M028127/1).

We would like to thank the YEAR Centre for their permission to use the Centre's grounds for the burial experiment and in particular Dr. Gareth Perry for his assistance in the burial and excavation of the experimental vessels.

For the morphological taxonomic verification of the *Salmo salar* specimens, we would like to thank Dr. Katrien Dierickx and for his assistance in figuring out Sankey diagrams we thank Dr. Jakob Hansen. Additionally, we would like to thank Dr. Claire Koenig for her advice on protein quantification.

Additionally, our gratitude goes to Patrick Gibbs and James Osborn from Heritage360, University of York, for access to their 3D scanning equipment and assistance in the acquisition and analysis of the 3D data.

Lastly, we would like to thank the two anonymous reviewers for their constructive comments, which helped improve the manuscript.

## Author contributions

**Conceptualization:** Joannes Adrianus Antonius Dekker.

**Formal analysis:** Joannes Adrianus Antonius Dekker.

**Funding acquisition:** Matthew Collins, Jessica Hendy.

**Investigation:** Joannes Adrianus Antonius Dekker.

**Methodology:** Joannes Adrianus Antonius Dekker, Richard Hagan.

**Project administration:** Joannes Adrianus Antonius Dekker, Matthew Collins, Jessica Hendy.

**Resources:** Richard Hagan, Matthew Collins, Jessica Hendy.

**Software:** Richard Hagan.

**Supervision:** Richard Hagan, Matthew Collins, Jessica Hendy.

**Validation:** Joannes Adrianus Antonius Dekker, Richard Hagan.

**Visualization:** Joannes Adrianus Antonius Dekker.

**Writing – original draft:** Joannes Adrianus Antonius Dekker.

**Writing – review & editing:** Joannes Adrianus Antonius Dekker, Matthew Collins, Jessica Hendy.

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
