## [Decision Letter · Decision Letter 0]

15 May 2025

PONE-D-25-11386
An unclean slate, discrepancies between food input and recovered protein signal from experimental foodcrusts
PLOS ONE

Dear Dr. Dekker,

Thank you for submitting your manuscript to PLOS ONE. After careful consideration, we feel that it has merit but does not fully meet PLOS ONE’s publication criteria as it currently stands. Therefore, we invite you to submit a revised version of the manuscript that addresses the points raised during the review process.

As you address the reviewers' comments, please pay particular attention to R1's request for more detail regarding direct / indirect heating, burial conditions at York Experimental Archaeological Research Centre, and potential contaminants; and R2's comments regarding accessibility. 

We look forward to receiving your revised manuscript.

Kind regards,

Raven Garvey, Ph.D.

Academic Editor

PLOS ONE

 [The research was funded by the Marie Skłodowska-Curie grant agreement No 956351  from the European Union’s Horizon 2020 research and innovation programme.]. 

Reviewers' comments:

Reviewer's Responses to Questions

**Comments to the Author**

1. Is the manuscript technically sound, and do the data support the conclusions?

Reviewer #1: Yes

Reviewer #2: Yes

2. Has the statistical analysis been performed appropriately and rigorously? 

Reviewer #1: Yes

Reviewer #2: Yes

3. Have the authors made all data underlying the findings in their manuscript fully available?

Reviewer #1: Yes

Reviewer #2: Yes

4. Is the manuscript presented in an intelligible fashion and written in standard English?

Reviewer #1: Yes

Reviewer #2: Yes

5. Review Comments to the Author

Reviewer #1: Proteomic analysis of food residues in archaeological ceramics is an emerging field which has the potential to provide us with valuable information regarding utilisation of resources and reconstruction of dietary pathways and food chains. However, progress in the field is hampered by the relative paucity of experimental archaeology studies in this field. As such, the present manuscript is a valuable contribution in the investigation of the relationship between the input resources and the subsequent formation of foodcrusts and the extraction of proteins from them and their interpretation to trace back the resources used.

The manuscript is scientifically sound, and the methodological techniques used are relevant and well-described. The conclusions drawn are broadly supported by the experiments described in the manuscript, and the results are discussed in an archaeological and interpretive context.

However, I have a few minor questions and concerns which I believe need to be addressed before the manuscript can be published.

Materials and Methods –

Can the authors discuss the possible reasons behind the formation of the foodcrust in the rim rather than any other region? I presume this is due to the possible evaporation in the region near the rim, but I will be interested in what the authors think about it.

I assume the cooking simulation experiment was carried using indirect heating, i.e., the ceramics were not placed directly in a heating element. If that ist he case, can the authors please mention and discuss why that approach was chosen rather than direct heating? To me, it appears more likely that archaeological pots would have been heated directly in an open flame, and as such a direct heat would have been a better approximation of the treatment undergone by foodstuff, and hence the formation of foodcrust more representative what likely happened in an archaeological context.

Can the authors please provide descriptions of the burial conditions (namely substrate/soil, approximate temperature, moisture content vs dryness of the substrate etc.) at York Experimental Archaeological Research Centre?

Protein data analysis

Can the authors please search the mass spectrometry data against a database of common contaminants like cRap? This will be helpful in identifying the prevalence of contaminant proteins and any possible misidentitication regarding them, which can be a potential explanantion for the protein A0A3Q1LL35, as the authors mention in line 463.

Discussion

Disentangling mixtures & taxonomic representativeness

Can the authors please discuss the proteins recovered after various cooking events in some greater detail by comparing and contrasting between the proteins recovered after various extraction, with specific focus on proteins recovered after the first cooking event? It appears to me that the first cooking event is perhaps most likely to estimate the conditions of archaeological ceramics- to me, it appears unlikely that the same pot was used for multiple cooking events without scrubbing off any visible charred residue. If that is the case, the residues after the first cooking event are probably most relevant for interpretation of archaeological data, and as such need to discussed in greater detail.

Can the authors please add a small paragraph comparing the nature/type of the various proteins recovered from the foodcrust with the type of proteins recovered in (i) other studies involving proteomics on archaeological samples where evidence of similar foodstuff processing has been recovered (ii) other experimental studies involving recovery of proteins from ceramic matrix and involving similar foodstuff?

The authors go into this in some detail re Evans et al. (line 501), but I believe a similar comparison involving additional studies (and comments on the differences in the types of proteins extracted) will be beneficial. This is particularly relevant since some prior studies have shown the recovery of muscle/blood proteins as evidence of animal processing, contrasting with the disappearance of the Bos taurus proteins here after burial. Is it possible that the nature of the substrate (foodcrust vs calcified deposits vs ceramic matrix) can affect the protein preservation and hence the interpretation of resource utilisation?

Reviewer #2: In this manuscript, Dekker et al. describe an experimental archaeological study to create foodcrusts in clay vessels and characterize their protein content over cooking and a variety of input conditions. This project is very well planned and executed by the authors. I'm impressed with the organization of this relatively complex set of experiments.

Line 88-89: "one lipid-rich, one protein-rich and one carbohydrate-rich to mix" Add material or food after carbohydrate-rich.

Line 182: How were the foodcrusts homogenized?

Line 234-237: For G-PTM-D, at minimum include the number of PTMs included and the PTM groups. If only specific ones were included, add them here or in the supplemental.

Line 240-242: Include the download dates of the Uniprot databases.

Figure 3: Unless the protein input is a continuous measurement, use factor for the scale instead of number.

Table 1: Include a column of the inputs for each cooking event. It may abe difficult to cross reference figure 1 with table 1 in the published version.

Figure 4: The lines between the boxes may not be clear if the figure is small at the final publication.

Line 333-338: Use the protein name instead of the Uniprot accession or put the accession in parentheses. It is hard to tell what protein it is from the accession alone.

Line 534 and 535: Should Fig. 6 be Fig. 7? This seems to be about the foodcrust size and not tissue specificity.

Line 538-539: "Suggesting that the presence of such a fat-rich food is not necessarily a driver of foodcrust formation" I think it may be worth connecting this result to the use of fats in reducing sticking in modern cooking.

Line 541: Confirm that this should be Fig. 5 and not another figure. It could be figure 2 or 7.

SI 5: Use a colorblind accessible gradient instead of the Red-Green one.

6. PLOS authors have the option to publish the peer review history of their article (what does this mean?). If published, this will include your full peer review and any attached files.

Reviewer #1: No

Reviewer #2: No

---

## [Author Response · Author response to Decision Letter 1]

15 Jul 2025

We would like to thank both reviewers for their constructive reviews.

In the document 'Response to reviewers' you will find a point-by-point response to each of the comments and we hope to have improved the manuscript to your liking.

---

## [Editor Report · Decision Letter 1]

29 Jul 2025

An unclean slate, discrepancies between food input and recovered protein signal from experimental foodcrusts

PONE-D-25-11386R1

Dear Dr. Dekker,

We’re pleased to inform you that your manuscript has been judged scientifically suitable for publication and will be formally accepted for publication once it meets all outstanding technical requirements.

Kind regards,

Raven Garvey, Ph.D.

Academic Editor

PLOS ONE
---

## [Editor Report · Acceptance letter]

PONE-D-25-11386R1

PLOS ONE

Dear Dr. Dekker,

I'm pleased to inform you that your manuscript has been deemed suitable for publication in PLOS ONE. Congratulations! Your manuscript is now being handed over to our production team.

Kind regards,

on behalf of

Dr Raven Garvey

Academic Editor

PLOS ONE